# Ultrawideband Terahertz Absorber with Dielectric Cylinders Loaded Patterned Graphene Structure

**DOI:** 10.3390/ma14216427

**Published:** 2021-10-26

**Authors:** Shuxiang Liu, Shufang Li

**Affiliations:** Beijing Key Laboratory of Network System Architecture and Convergence, Beijing University of Posts and Telecommunications, Beijing 100876, China; shaniya@yeah.net

**Keywords:** fabry-perot cavity, graphene, broadband absorber, plasmon resonance

## Abstract

In this paper, we theoretically designed and numerically analyzed an ultrabroadband meta-absorber with near unity absorptivity that works in terahertz spectrum. A wideband meta-absorber composed of bilayer patterned graphene and dielectric cylinder array with high symmetry was proposed. The wideband absorption mechanism benefited from two aspects. The first one was enhanced surface plasmons based on bilayer patterned graphene. And the second one was the coupling of continuous resonant modes within Fabry-Perot cavities to the enhanced surface plasmons in the graphene. An ultrawide bandwidth with absorptivity over 90% were obtained from 3.2 THz to 9.4 THz. Simulated results showed that the proposed ultra-wideband absorbing structure also possessed high performance of polarization independence, flexible tunability, large incident angle insensitivity, and compact fabrication.

## 1. Introduction

Electromagnetic wave absorber is a device that converts electromagnetic wave into ohmic heat or other forms of energy by interacting with the electric field or magnetic field of the incident electromagnetic wave [1]. Electromagnetic wave absorbers were early used in military fields to achieve aircraft stealth or radar detection. Especially, with the rich frequency resources of terahertz and the ability to transmit nanosecond and picosecond pulses at thousands of frequencies, terahertz absorber has wider application range beyond the existing stealth technology.

Traditionally, the absorption was realized using absorbing materials, by absorbing electromagnetic energy efficiently and reducing the reflection. Since Landy first proposed the metamaterial perfect absorber in 2008 [2], the electromagnetic wave can be controlled artificially by adjusting the shapes and properties of metamaterials to achieve high absorption in sub wavelength structure. Then the metamaterial absorber has gradually extended to dual band, triple band, multi band, broadband and tunable absorption [3,4,5,6]. For multi-band absorbers, they behave with strong frequency selectivity. However, in some other applications where broadband sensing and detecting are needed, broadband absorbers are of great importance. Among them, broadband and tunable absorbers are attracting considerable attention. Tunable broadband absorber can be widely used in thermal radiation [7] detection and terahertz stealth [8,9]. The increasingly complex electromagnetic application environment needs active tunable electromagnetic metamaterial absorber with the change of different active demands [10,11,12].

Many tunable absorbers with materials consisting graphene [13,14], black phosphorus [15,16], vanadium dioxide [17] and MoS2 [18] have entered the research field. Especially, graphene becomes the excellent one with the advantage of the highest carrier mobility and mature controllability of mechanical properties. Especially with the maturity of graphene preparation technology and the reduction of production cost, there are more opportunities for graphene to apply to more functional materials and applications [19,20]. Through changing the electrostatic doping or chemical doping [21], the complex conductivity of the graphene can be adjusted accordingly. What’s more, graphene can interact with electromagnetic wave to produce surface plasmons under certain conditions, enabling strong coupling between plasmon and incident electromagnetic wave and enhancing the energy absorption in infrared and terahertz [22,23].

However, the absorption bandwidth relied on resonances of surface plasmon polaritons of graphene is too narrow to satisfy certain potential applications [24,25]. Much effort has been done to achieve broadband absorption based on graphene, multilayers of graphene with different biasing voltages are adopted to improve the absorption bandwidth [26,27]. However, stacking structure with multilayers of graphene increases the difficulty of manufacture and makes it hard to match desired results in the experiments. Alternatively, merging multiple resonators in one cell with biperiodic or multi-periodic patterned graphene can also achieve wide absorption bandwidth [28,29]. The later approach has limited bandwidth improvement space whether from the essence of resonance or from the limitation of structural space. Recently, heavily doped silicon grating terahertz absorbers attracted extensive attention for achieving high absorption within a wide spectrum in the terahertz regime [30,31]. However, the profile of the grating absorbers appears bulky and makes it hard to integrate. Another strategy is combining grating structure and graphene [32,33,34], which can broaden the absorption spectrum, at the same time, simplify the device fabrication and realize miniaturization. In [32,33,34], broadband absorption is achieved using the periodical array of dielectric bricks or elliptic dielectric cylinders coated with single-layered and nonstructured graphene. However, the profile of the absorbing structure is still large to integrate with the existing equipment and systems.

In this letter, we propose a polarization insensitive broadband meta-absorber based on patterned graphene in terahertz spectrum. We utilize the enhanced surface plasmons with dielectric cylinders loaded patterned bilayer graphene structure and merge varied resonances to realize the increased absorption within an ultra-wide bandwidth. Due to simultaneously generated electric and magnetic resonances of both plasmon resonance and Fabry-Perot (FP) resonance, the proposed broadband absorber can have tunable absorption spectrum from 3.2 THz till 9.4 THz, with an absorption rate over 90% under normal incidence. The wideband absorption mechanism benefited from two aspects. The first one is enhanced surface plasmons based on bilayer patterned graphene. And the second one is the coupling of continuous resonant modes within Fabry-Perot cavities to the enhanced surface plasmons in the graphene. The multiple plasmon resonances and FP cavity resonances inoculate a simple way to build ultra-wideband perfect absorption with polarization independence, large incident angle insensitivity, and compact fabrication.

## 2. Structure Configuration and Absorption Principle

The schematic unit cell of the proposed absorption structure is depicted in Figure 1. In top-down order, the unit consists of one dielectric cylinder, bilayer patterned graphene and a metallic ground at the bottom with the unit period of P=14 μm. On the top, the dielectric cylinder is a lightly doped silicon ($\epsilon_{r}=11.8,tan\sigma =0.05$) with the height $H_{3}=1\;$μm and the radius R=1 μm. For the convenience of process, the bilayer graphene is set as the same pattern. The patterned bilayer graphene separated with dielectric. Figure 1b gives the details of the graphene ring and disk in the unit cell. It consists of one outer ring with outer radius $R_{1}=6.3\;$μm and innner radius radius $R_{2}=3.3\;$μm, the radius of the inner disk is $R_{3}=2.9\;$μm, which is separated with the ring by a gap g=0.4 μm. The dielectric layer between the bilayer graphene is the same with the substrate under the second graphene layer, which comprises the dielectric TOPAS. The thickness are set as $H_{1}=6.7\;$μm and $H_{2}=1\;$μm, respectively, as shown in Figure 1c. TOPAS is selected as the substrate for the reason of low absorption and it’s refractive index is between 1.52 and 1.53 in terahertz band. A permittivity ϵ = 2.60 and loss tangent = 0.04 are used to model the TOPAS film. At the bottom layer, Au is selected as the ground plate with a thickness of 200 nm, whose relative permittivity is described by Drude model as [35]: $\epsilon \left(\omega \right)=\epsilon_{\infty }-\omega_{p}^{2}/\left(\omega^{2}+i\rho \omega \right)$, with $\epsilon_{\infty }=1.0$, $\omega_{p}=4.35\pi \times 10^{15}$ s${}^{-1}$ and $\rho =8.17\times 10^{13}$ s${}^{-1}$.

In the above configuration, the periodic silicon cylinder arrays loaded with patterned graphene can also supply local resonant reflection, forming an asymmetric FP cavity. When the incoming electromagnetic wave propagates in the multilayer medium, the electric field and magnetic field will be modulated, and the propagation state of the electromagnetic wave will change complex. Different from the traditional medium, graphene is a monolayer of carbon with a specific conductivity governed by the Kubo formula [36]. In the terahertz range, the Kubo conductivity included the inter-band and intra-band transition contributions. The contribution of the interband electron transfer is negligible compared with the intra-band electron transfer [36], which can be expressed as:(1)$$\begin{array}{cc} \delta_{g}=\delta_{intra}\left(\omega ,\mu_{c},\Gamma ,T\right) & \approx \frac{-je^{2}k_{B}T}{\pi \hbar^{2}\left(\omega -j2\Gamma \right)}\left[\frac{\mu_{c}}{k_{B}T}+2ln\left(e^{-\mu_{c}/\left(k_{B}T\right)}+1\right)\right]. \end{array}$$
where $k_{B}$ is the Boltzmann’s constant, *T* is the temperature in Kelvin, e is the electron charge, and $\mu_{c}$ is the chemical potential. Moreover, ω is the angular frequency, *ℏ* is the reduced Planck’s constant, τ is the electron relaxation time, and Γ=1/(2τ) represents the electron scattering rate. Here assuming T=300 K and τ=0.09 ps.

Figure 2 shows how the real and imaginary parts of graphene conductivity are dynamically changed as the Fermi level varies from 0.1e V to 0.9 eV. Thus the propagation state of the electromagnetic wave will be controlled accordingly. In lower terahertz band, the highly doped energy of graphene($\mu_{c}>\hbar \omega$) can support plasmons propagating along the surface with the wave vector as [37,38]: (2)$$\begin{array}{c} k_{GSP}\left(\omega \right)=\frac{\pi \hbar^{2}}{e^{2}\mu_{c}}\epsilon_{0}\left(\epsilon_{r_{1}}+\epsilon_{r_{2}}\right)\omega \left(\omega +j\tau^{-1}\right) \end{array}$$
here, $\epsilon_{0}$ is the permittivity of vacuum, $\epsilon_{r_{1}},\epsilon_{r_{2}}$ are the effective relative permitivity of substrates above and below the graphene film. The result of $k_{GSP}$ needs to be adjusted to eliminate difference with the wave vector in air, to achieve near-unity absorption within a wide band. Patterned graphene can indirectly change the property of graphene plasmons. Attaching periodic silicon cylinders with high permitivity to the top will ameliorate the mismatch further. Then, when the incident wave with electric field linearly polarized in y-axis direction, the surface plasmons on graphene will be excited, which propagate along the direction of the electric field polarization. And the silicon cylinders can be regarded as the combination of infinite isosceles trapezoids segments along the y-axis, and the two faces opposite in the x direction behave as the two reflective mirror faces of one Fabry-Perot cavity, as shown in Figure 1c. Each of the Fabry-Perot cavity lengths $L_{FP}\left(y\right)$ can be approximated as:(3)$$\begin{array}{c} L_{FP}\left(y\right)=2\sqrt{1-\frac{y^{2}}{R^{2}}} \end{array}$$

The surface plasmons of graphene can be achieved when the wave vector matching condition is met as:(4)$$\begin{array}{c} Re\left(k_{GSP}\right)\cdot 2L_{FP}\left(y\right)=2N\pi +2\phi ,N=1,2,3,\ldots \end{array}$$
where $Re(k_{GSP})$ is the real part of wave vector, N relates to the order of resonance mode, ϕ is the phase of reflection at the mirror face of each segments. Then if we combine Equations (2)–(4), we can obtain the angular frequency for each order of plasmon mode as:(5)$$\begin{array}{c} \omega_{N}=\frac{e}{\hbar }\sqrt{\frac{\mu_{c}\left(N\pi +\phi \right)}{2\pi \epsilon_{0}\left(\epsilon_{r_{1}}+\epsilon_{r_{2}}\right)\sqrt{R^{2}-y^{2}}}} \end{array}$$

Then we can conclude that our structure can produce multiple discrete plasmon resonances, what’s more, for each mode, a continuous resonances exist for the reason of continuously variable y. Then, the multi-band continuous resonances can be adjusted through parameter scanning to locate within the spectrum of two absorption peaks caused by the patterned graphene. In the silicon cylinders, we define β as the angle between two mirrors of Fabry-Perot cavities, which has a great influence on the resonance and beta changes gradually around zero to ensure strong resonances.
(6)$$\begin{array}{c} \beta \left(y\right)=\pi -2tan^{-1}\left(\sqrt{\frac{R^{2}}{y^{2}}-1}\right) \end{array}$$

Equation (Equation 6) implies that more resonances in the FP cavity can be achieved under the premise of y and angle β close to zero. At the same time, through adjusting *R*, the plasmon resonance can be adjusted within the expected spectrum. Then, we will have a wideband property with near-unity absorption under certain conditions.

## 3. Simulation Verification and Discussion

When the uniform electromagnetic wave is incident into the multilayer medium, part of the energy is reflected and part is transmitted. The final absorption rate in this paper can be expressed as A(ω)=1−R(ω)−T(ω) with total reflection given by $R\left(\omega \right)=|S_{11}{\left(\omega \right)|}^{2}$ and total transmission given by $T\left(\omega \right)=|S_{21}{\left(\omega \right)|}^{2}$, where $S_{11}$ and $S_{21}$ are frequency dependent return losses and transmission losses, respectively. In the proposed structure, the total transmission T(ω) is equal to 0 due to skin effect. The thickness of the gold on the back (n=200 nm) is greater than its skin depth in terahertz band, leading to the final absorption rate represented as $A\left(\omega \right)=1-R\left(\omega \right)=1-|S_{11}{|}^{2}$.

Theoretical research is developed using the commercially available finite element simulator HFSS. To asure the computing accuracy and high efficiency, the term Maximum Number is set to 15 and the term Maximum Delta S is set to 0.01. In the simulation, the bilayer graphene are all set as impedance boundary with resistance: 1/coff/τ, and reactance: 2πFreq/coff. Where, $coff=e_{0}^{2}k_{B}T/\pi /\hbar^{2}\left(\mu_{c}/k_{B}/T+2ln\left(e^{\left(-\mu_{c}/k_{B}/T\right)}+1\right)\right)$. For numerically computing the scattered fields and the scattering parameters of the proposed absorber, the periodicity of the meta-absorber is realized through the Master-Slave boundary conditions, as shown in Figure 3a. The master–slave boundary conditions are set with XOZ and YOZ face. Floquet Ports are set as the excitation located at the two-sides of z-direction. The TE-polarized or TM-polarized electromagnetic wave is then exerted to define the detailed excitation. In our proposed structure, the TE mode means the component of the electric field parallel to the surface of the proposed structure (XOY plane) and the the TM mode means the component of the magnetic field parallel to the surface of the proposed structure. As shown in Figure 3b, the obtained absorption of the proposed structure is over 90% from 3.2 THz to 9.4 THz.

Before analyzing the absorption mechanism of the whole structure, the adoption of bilayer patterned graphene is clarified below. The comparison of normalized absorption spectra of three kinds of absorber with monolayer graphene, bilayer graphene and tri-layer graphene are plotted respectively, as shown in Figure 3b. Obviously, with adhesion of graphene layer mounted on a thin dielectric layer, the absorption rate for two absorption peaks are enhanced. And we can see that bilayer graphene is enough for improving the peaks absorption rate from 83% to 95.8%, and from 85% to 96.9%.

Then, no matter how to tune the loading voltage, the absorption in a wide band can not be improved only based on the bilayer structure. Once the periodic silicon cylinders with high permitivity are attached to the top, as mentioned above, the simulation result shows that we indeed have improved the absorption rate within the wide spectrum, as shown in Figure 3b. The high absorbtivity over 90% were obtained from 3.2 THz to 9.4 THz. The final result is achieved with the adopted dielectric cylinders loaded bilayer patterned graphene.

To further validate the absorption mechanism, simulated electric field distribution, surface current density of the first graphene layer, the second graphene layer and the backed Au ground and the profile magnetic field distribution of structure at 3.8 THz, 6 THz and 8.2 THz are presented in Figure 4 and Figure 5, respectively.

In 3.6 THz, strong electric filed is concentrated on both the edge of the graphene and the center of the graphene along the direction of the incident wave. When the incident wave is polarized in the direction of y-axis, the electric resonance is along y-axis direction. The bottom graphene layer has the same resonant modes with the upper graphene layer with a comparable higher electric field maximum amplitude at 3.6 THz, as shown in Figure 4a,d. Combined with the opposite direction of the surface current on the bilayer graphene and the metallic ground at the bottom, as shown in Figure 4, we can obtain a conventional absorption mechanism with magnetic dipole response at 3.6 THz. And ohmic loss based on plasmon resonance is the cardinal loss for the first absorption peak of the electromagnetic energy.

For the second absorption peak of 6 THz, strong electric field appears in the edge of the graphene and the junction of graphene and cylindrical dielectric, as shown in Figure 4. The surface currents on the bilayer graphene and the metallic ground at the bottom are syntropic, which means the absorption is mainly caused by the surface plasmons, as shown in Figure 5b,e,h. We can find that the graphene plasmon resonances (GPRs) are extended because of the coupling between the graphene and the cylindrical. To intuitively verify the absorption mechanism, distributions of electric field at the distance 50 nm above the interface between upper layer of patterned graphene and dielectric cylinder array for nine sample frequencies are presented in Figure 6. It is obvious that different absorption frequencies correspond to different electric field patterns. For Figure 6a, on the circle of R, there is a first-order graphene plasmon resonance. For Figure 6b, we can see that the field patterns consist of second order mode with the combination of graphene plasmon resonances of the inner circle. Different absorption frequencies correspond to different FP cavity lengths along y direction in the dielectric cylinder. Similarly, the high order continuous absorption band can be formed, as shown in Figure 6c–i, however, it is hard to discriminate each mode for the reason of overlapping resonances. From Figure 4h,i, it is obvious that the field is mainly located at the junction of the dielectric cylinder and the patterned graphene, revealing that the absorption mechanism behind correlates with GPRs highly. These phenomenons further verify the theoretical predictions before. Section magnetic field of the structure in XOZ plane is also depicted in Figure 5k. It is obvious that the magnetic field is mainly confined within the bottom of silicon cylinders and the cavity between bilayer graphene. Then we can conclude that the high absorption is propelled by the magnetic field and the electric field confines of the plasmon resonance. When it comes to 8.2 THz, the strong electric plasma resonances shift mainly to the junction of the graphene and cylindrical dielectric, validating the absorption mechanism of multiband continuous graphene plasmon resonances for our previous analysis.

Then in order to give the verification of the mechanism derived above further, the main parameters that affect the main absorption peaks are analyzed and presented in Figure 7. The first absorption peak can be individually controlled when the gap g varies while the other parameters keep unchanged. At the same time, the other two absorption peaks remain almost unchanged, as shown in Figure 7a. What’s more, as shown in Figure 7d, the first absorption peak becomes weaker as the dielectric spacer distance $H_{1}$ decreases, which is rooted in the direct relation to the localized surface plasmon polaritons. However, at the same time, as $H_{1}$ decreases, hybrid localized surface plasmon polaritons and delocalized surface plasmon polaritons bring a relative wider absorption bandwidth. And when the radius R of additional silica cylinders is larger than the radius of inner ring of graphene or stays near the radius of the outer ring radius of the graphene, the effect of silicon cylinders and patterned graphene enables an improvement of the absorption bandwidth further. As shown in Figure 7b, here we define $R=R_{3}+r$, and r changes from −1.0 μm to 2.0 μm. When r≥1.6 μm, the third plasmon absorption peak disappears, and when r≤0.4 μm, the second absorption peak becomes weak. when r is near 1.1 μm, the bandwidth with high absorption is the optimal. Similarly, the height of the cylinder $H_{3}$ has tight relation with the second absorption peak. The greater $H_{3}$ makes the second absorption peak closer to the first absorption peak, and considering the low profile of the structure, $H_{3}$ values at 2 μm, as shown in Figure 7c.

Even if the structural parameters are fixed, the tunability of the absorption is still achievable benefited from the controllable eletromagnetic characteristics of graphene. Through adjusting the biasing voltage from 0.1 eV to 0.97 eV, as shown in Figure 7e, the whole absorption bandwidth can be controlled accordingly. The optimal absorption bandwidth can be achieved when the applied voltage is high enough. When the voltage decreases to a low value of 0.1 eV, the localized surface plasmon polaritons decrease. However, there is still a remaining absorption peak caused by periodic arrays of silicon cylinders loaded with patterned graphene.

Next, we will investigate the property of polarization independence under different incident angles. In the above mentioned analysis, we discussed the absorption characteristics only under normal incidence scenario with angle $\phi =0^{\circ }$ and $theta=0^{\circ }$. The polarization angle ϕ is defined as included angle between *x*-axis and electric vector of plane wave. We have investigated the absorption consistency of variational trend under the changes of the polarization angle ranging from $0^{\circ }$ to $90^{\circ }$ in steps of $10^{\circ }$, at varying incidence angle θ from $0^{\circ }$ to $60^{\circ }$, with a step width of $20^{\circ }$. As shown in Figure 8a and Figure 9, the absorption property is consistant under various polarization angles at certain incident angle. In other words, the proposed structure has high polarization independence no matter how the angle of incident wave changes, especially applying to terahertz applications with non-polarized source.

Besides, the absorption spectrum under varied oblique incidence in both the TE and TM modes are studied, as depicted in Figure 8b,c, respectively. In the simulation, the incidence angles vary from $0^{\circ }$ to $80^{\circ }$ with the step width of $10^{\circ }$. It can be presented that the first peak absorption decreases as θ increases, which keeps larger than 60% up to $60^{\circ }$ incidence angles for both the TE mode and the TM mode, respectively. However, the absorption rate of the second and third absorption peaks can still keep larger than 80% up to $70^{\circ }$, presenting excellent performance of independence for incidence angle in both the TE and TM modes.

Finally, we obtain a polarization insensitive and wide angle independence broadband meta-absorber with absorption larger than 90% from 3.2 THz till 9.4 THz. Enhanced surface plasmons with dielectric cylinders loaded patterned bilayer graphene structure are utilized to achieve broad absorption bandwidth. Through analyzing three absorption peaks, the absorbing mechanisms behind is presented by means of electric field and magnetic field diagram. Flexible frequency and absorbance adjustability are given through tuning the biasing voltage of graphene. Despite the absorption property under normal incident wave excitation, the characteristics of polarization independence under various incidence angles and wide angle insensitivity are also investigated, the structure can still keep absorption rate larger than 60% with incidence angle of $60^{\circ }$ from 3 THz to 11 THz, and absorption rate larger than 70% with incidence angle of $70^{\circ }$ from 6.3 THz to 10.9 THz. The comparison with some previous researches is presented in Table 1, proving to be an ultra-wideband perfect absorption with exellent property of polarization independence, large incident angle insensitivity, and compact fabrication simultaneously.

## 4. Conclusions

In summary, a bilayer graphene-based absorption structure with Au cylinder arrays on parylene substrate is proposed and numerically simulated at THz wavelengths. We use the patterned bilayer graphene to produce localized surface plasmon polaritons and delocalized surface plasmon polaritons to obtain dual frequency absorption, then additional silicon cylinder arrays on the top of the patterned graphene produce a series of discrete plasmon resonances located within the dual frequency interval, at the same time, enhancing the absorption rate through improving the matching condition. Ultra-wide bandwidth is realized when the guided surface plasmon resonance is enhanced by tuning the parameter of silicon cylinders, promoting field confinement on the surface of the structure. The absorption rate and bandwidth of the structure can be independently tuned by varying the chemical potential of biasing voltage on both of the graphene layers. Simulation results demonstrate that the absorption of the proposed structure can be as high as more than 90% from 3.2 THz to 9.4 THz under normal incidence. what’s more, the proposed structure also possesses the excellent property of polarization independence and wide angle insensitivity for both TE and TM polarization. The comparison of demanding properties with some previous researches proves that our proposed tunable ultra-broadband absorber fully deserves to be an excellent and functional candidate in sensing, thermal radiation, detection, terahertz stealth and adsorbents, etc.

## Figures and Tables

**Figure 1 materials-14-06427-f001:**
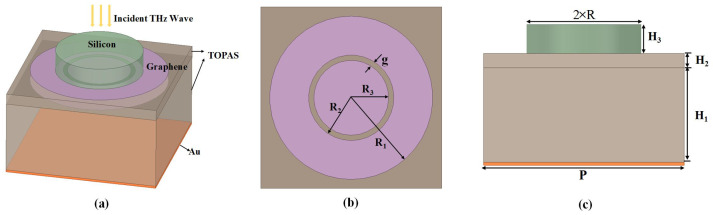
(**a**) Schematic of the proposed absorber. (**b**) Top view of the unit cell of patterned graphene, (**c**) Side view of the unit cell.

**Figure 2 materials-14-06427-f002:**
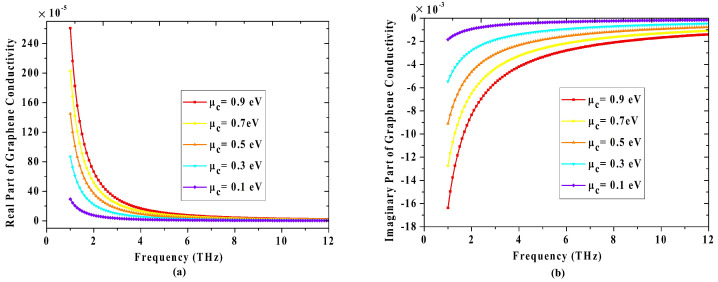
The (**a**) real part and (**b**) imaginary part of graphene conductivity with the change of the Fermi level.

**Figure 3 materials-14-06427-f003:**
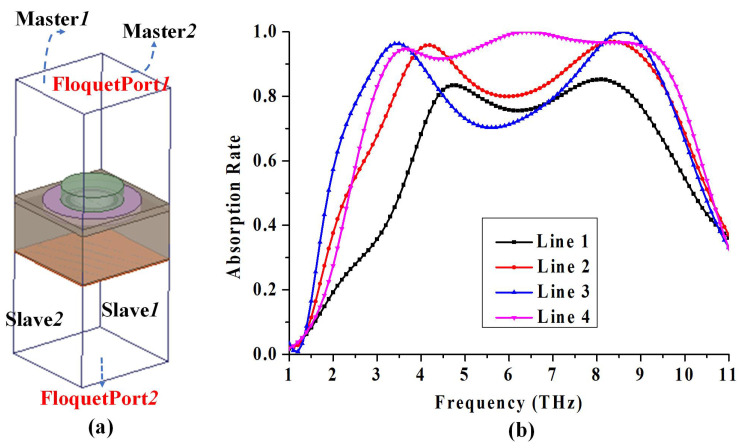
(**a**) Boundary and Excitation setting for the unit cell of the proposed absorber, (**b**) The simulated frequency responses for Line 1: with one patterned graphene layer, Line 2: with bilayer patterned graphene, Line 3: with trilayer patterned graphene, Line 4: with dielectric cylinder and bilayer patterned graphene.

**Figure 4 materials-14-06427-f004:**
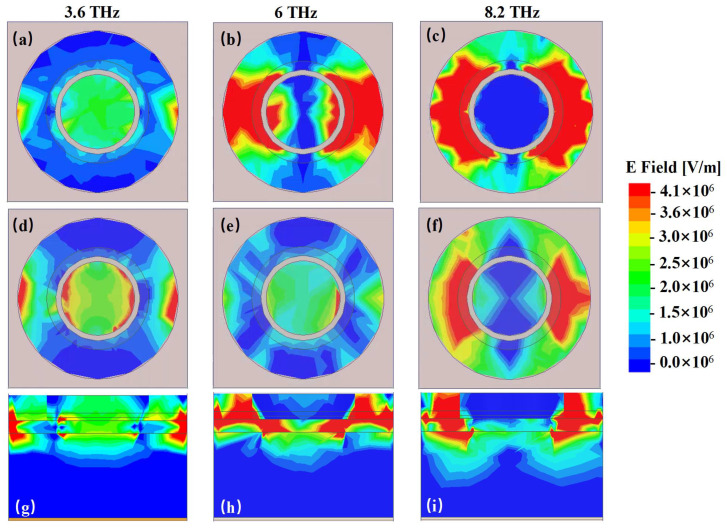
The simulated electric field distribution of the top graphene layer at (**a**) 3.6 THz, (**b**) 6 THz, (**c**) 8.2 THz and bottom graphene layer at (**d**) 3.6 THz, (**e**) 6THz, (**f**) 8.2 THz and side view of electric field at (**g**) 3.6 THz, (**h**) 6 THz, (**i**) 8.2 THz.

**Figure 5 materials-14-06427-f005:**
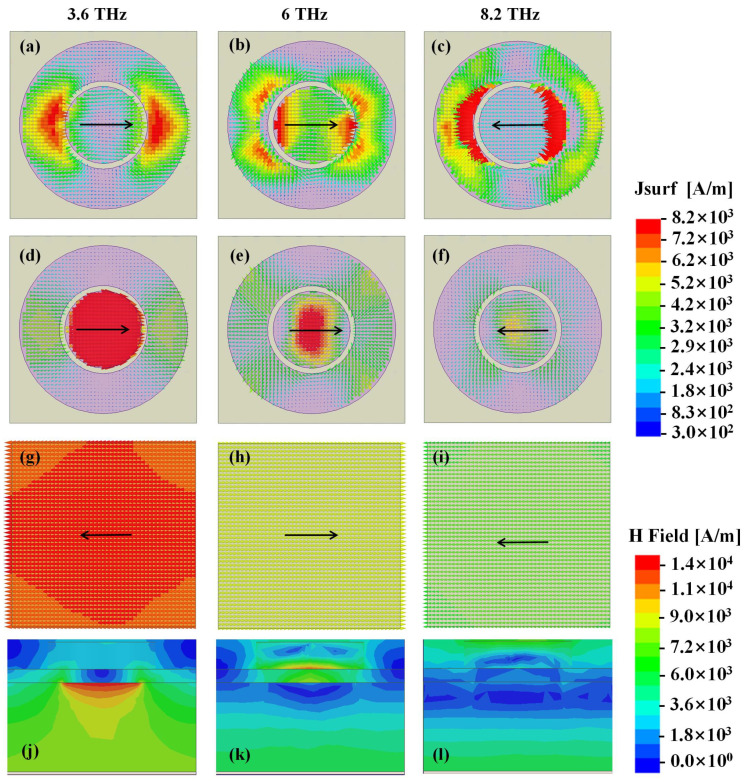
Simulated surface current density of the top graphene layer at (**a**) 3.6 THz, (**b**) 6 THz, (**c**) 8.2 THz and bottom graphene layer at (**d**) 3.6 THz, (**e**) 6 THz, (**f**) 8.2 THz and backed ground layer at (**g**) 3.6 THz, (**h**) 6THz, (**i**) 8.2 THz and the profile of magnetic field at (**j**) 3.6 THz, (**k**) 6 THz, (**l**) 8.2 THz.

**Figure 6 materials-14-06427-f006:**
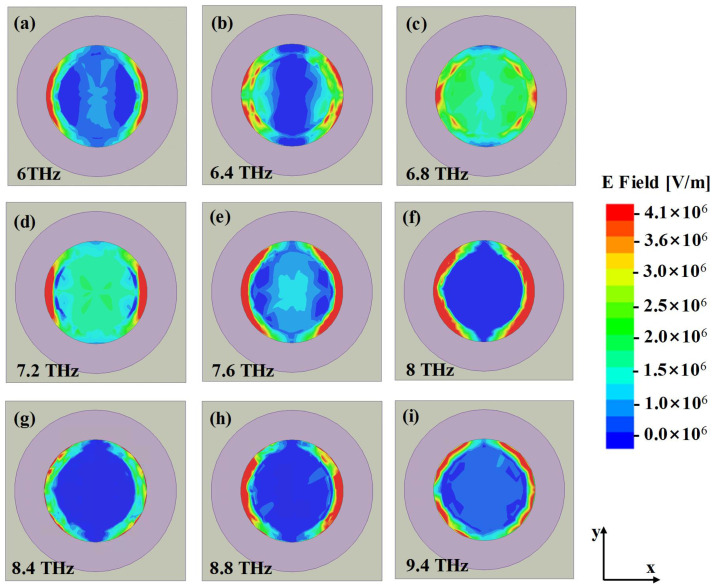
Typical distributions of the electric field at the distance 50 nm above the interface between the upper graphene and the silicon cylinder at (**a**) 6 THz, (**b**) 6.4 THz, (**c**) 6.8 THz, (**d**) 7.2 THz, (**e**) 7.6 THz, (**f**) 8 THz, (**g**) 8.4 THz, (**h**) 8.8 THz, (**i**) 9.4 THz.

**Figure 7 materials-14-06427-f007:**
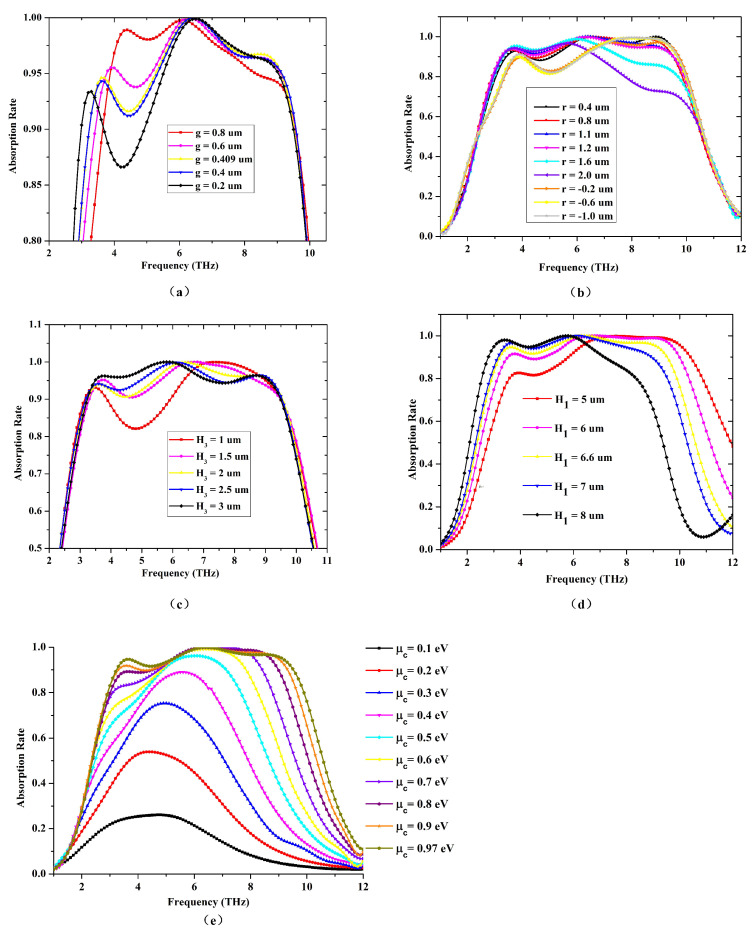
The simulated absorption rate with the change of (**a**) g, (**b**) r, (**c**) $H_{3}$, (**d**) $H_{1}$ and (**e**) the Fermi level $\mu_{c}$.

**Figure 8 materials-14-06427-f008:**
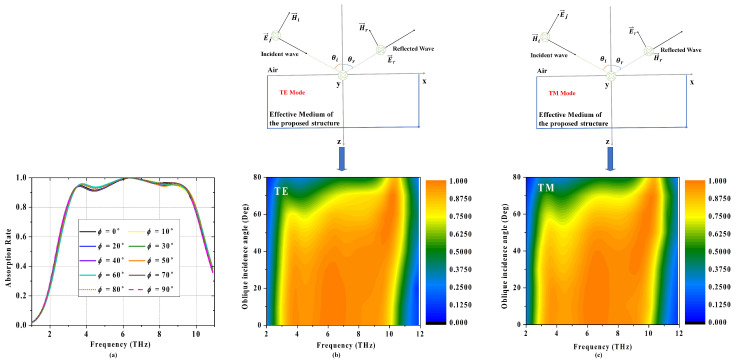
(**a**) Absorption rate under various polarization angle. (**b**) Absorption contour map of the absorber under various incident angles of (**b**) TE mode and (**c**) TM mode.

**Figure 9 materials-14-06427-f009:**
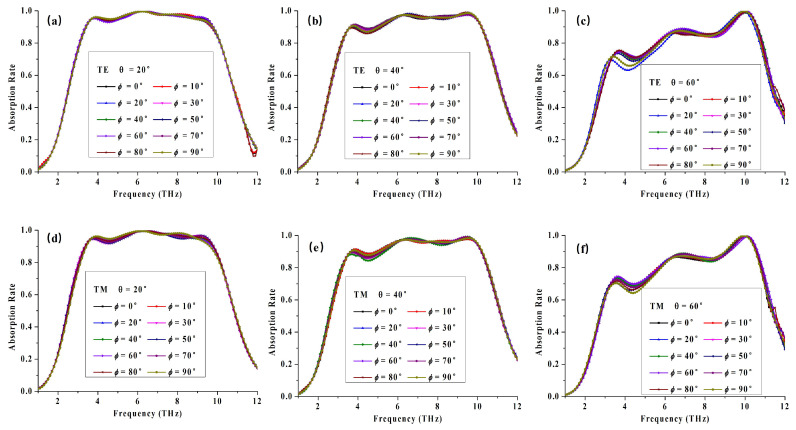
Simulated absorption spectra of the absorber for different polarization angles for TE mode under incident angle of (**a**) $20^{\circ }$, (**b**) $40^{\circ }$, (**c**) $60^{\circ }$, and for TM mode under incident angle of (**d**) $20^{\circ }$, (**e**) $40^{\circ }$, (**f**) $60^{\circ }$.

**Table 1 materials-14-06427-t001:** Comparison between our proposed results and some previous researches.

Ref (Year)	Bandwidth (THz)	RAB (≥90%)	Angular Stability	Profile (um)	Tunability
30 (2016)	0.75–2.41	105	unknown	120	N
25 (2016)	1.666–2.562	42	60	10.2	N
24 (2017)	1.72–4.42	87.9	60	15	Y
33 (2018)	1.6–3.1	65	60	28	Y
35 (2019)	1.52–3.2	68	unknown	39	Y
29 (2021)	1.26–1.5848	20.5	60	28	Y
This Paper	3.2–9.4	98.4	70	8.7	Y

## Data Availability

Data is contained within the article.

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
