# Peer review of "Ultrawideband Terahertz Absorber with Dielectric Cylinders Loaded Patterned Graphene Structure"

_materials, 2021, doi:10.3390/ma14216427_

Round 1

Reviewer 1 Report

The submitted manuscript reports the design and simulations of a broadband terahertz absorber made of patterned graphene. Even though the paper has some practical appeal, it looks like a rather incremental addition to the already existing literature on a similar subject. Because the authors do not provide a convincing discussion of the state of the art, the scientific novelty of the paper seems to be rather weak.  The authors do not mention any paper on the design and performance of the graphene-based terahertz absorber published during last two years (and there are many papers on the subject, just single paper (Kai-Da Xu, Jianxing Li, Anxue Zhang, and Qiang Chen, "Tunable multi-band terahertz absorber using a single-layer square graphene ring structure with T-shaped graphene strips," Opt. Express 28, 11482-11492 (2020)) gained nearly 20 citations. Given an already existing substantial literature on the subject, I cannot endorse the submitted paper. The performed simulations should be followed by the feasibility analysis. The authors should show that the proposed design can be realized experimentally. The choice of materials should be justified. The range of allowed intensities of the incident light should be discussed (because of very large values of the local electric field it must be a material limit on the maximum intensity of the incident light).

Reviewer 2 Report

This manuscript describes a theoretical study of a graphene/dielectric microstructure to generate a broadband terahertz absorber. There are two corrections that are essential:

  • Is the disk on the top of the assembly silicon, silica, or silicone? One is a semiconductor, the other two are insulators. Only two can be vapor deposited, while all can be lithographically etched. The sloppy mixing of the terms must be fixed. My guess is that the disk is silica (SiOx), but there should be no ambiguity.
  • In Figure 1c, H1 looks different than H2, yet in the text H1 = H2. The visual and the text should be consistent.

The language requires review of a native English speaker.

The microstructure is carefully analyzed in isolation. In any real world application, one would grow an array of identical devices. The interaction of adjacent devices should be included in data analysis. While the disks are widely separated compare to R2-R3, there will be some coupling, particularly if incident radiation isn’t normal to the disk surface.

Reviewer 3 Report

Dear Authors

The manuscript describes theoretically designed and numerically analyzed an ultra-broadband meta-absorber with near-unity absorptivity that works in the terahertz spectrum. This manuscript can be accepted after major revision.

The following suggestion and comments should be taken:

  1. The overall English needs to be improved. Please seek guidance from a native English speaker if possible (commas, plural form,  "the" "a", and others could be corrected).
  2. Line 32-38 The introduction section needs enhancement. Please add some new information about graphene: methods of obtaining, potential applications; easy modifications with other carbons, properties that make it so interesting. Please cite (1) J. Mater. Chem. C, 2021,9, 6722-6748.  DOI https://doi.org/10.1039/D1TC01316E; (2) Materials 2021, 14(9), 2448; https://doi.org/10.3390/ma14092448; (3) Appl. Sci. 2020, 10, 1753; doi:10.3390/app10051753.
  3. Figure 2, please correct to better quality.
  4. Could the authors include the standard deviation of the simulated methods?
  5. Could authors write the difference between TM and TE mode?
  6. In conclusion please write 1-2 sentences potential applications of such materials.

Round 2

Reviewer 1 Report

The authors addressed the majority of the received comments. 

Reviewer 3 Report

Accept in present form